# Ocular dominance-dependent binocular combination of monocular neuronal responses in macaque V1

**Sheng-Hui Zhang**[1,2†], **Xing-Nan Zhao**[1,2†], **Dan-Qing Jiang**[1,2], **Shi-Ming Tang**[2,3,4*], **Cong Yu**[1,4*]

[1]School of Psychological and Cognitive Sciences, Peking University, Beijing, China; [2]PKU-Tsinghua Center for Life Sciences, Peking University, Beijing, China; [3]School of Life Sciences, Peking University, Beijing, China; [4]IDG-McGovern Institute for Brain Research, Peking University, Beijing, China

**\*For correspondence:**
tangshm@pku.edu.cn (SMT);
yucong@pku.edu.cn (CY)

†These authors contributed equally to this work

**Competing interest:** The authors declare that no competing interests exist.

**Abstract** Primates rely on two eyes to perceive depth, while maintaining stable vision when either one eye or both eyes are open. Although psychophysical and modeling studies have investigated how monocular signals are combined to form binocular vision, the underlying neuronal mechanisms, particularly in V1 where most neurons exhibit binocularity with varying eye preferences, remain poorly understood. Here, we used two-photon calcium imaging to compare the monocular and binocular responses of thousands of simultaneously recorded V1 superficial-layer neurons in three awake macaques. During monocular stimulation, neurons preferring the stimulated eye exhibited significantly stronger responses compared to those preferring both eyes. However, during binocular stimulation, the responses of neurons preferring either eye were suppressed on the average, while those preferring both eyes were enhanced, resulting in similar neuronal responses irrespective of their eye preferences, and an overall response level similar to that with monocular viewing. A neuronally realistic model of binocular combination, which incorporates ocular dominance-dependent divisive interocular inhibition and binocular summation, is proposed to account for these findings.

## eLife assessment

Overall, the reviewers found the significance of the work **valuable** to the field of visual neuroscience, particularly given the large data set and strength of the method used that allowed for spatial analysis of neuronal responses in macaque V1. The evidence was deemed **compelling**, owing in part to the consistency of responses across animals and the fitness of modeling. The authors have addressed the major comments from reviewers and improved the manuscript through relation to prior literature and addressing specific limitations of the method used.

## Introduction

Human and non-human primates often rely on binocular disparity, which refers to the differences between the retinal images of the two eyes, to perceive depth (stereopsis). In the meantime, the brain maintains a stable perception of the visual world when either one eye or both eyes are open and the light entering the brain is either halved or doubled. Much has been known regarding the neural foundations of stereopsis (*Barlow et al., 1967*; *Henriksen et al., 2016*; *Parker et al., 2016*; *Welchman, 2016*; *Read, 2021*). Nevertheless, whether and how the neurons respond differently to monocular and binocular stimulations is less studied (*Poggio and Fischer, 1977*; *Prince et al., 2002*; *Dougherty*

*et al., 2019*; *Mitchell et al., 2022*). Adding to the complexity is the fact that many V1 neurons, although responding to stimulations from both eyes, have various degrees of eye preferences (*Hubel and Wiesel, 1962*; *Shatz and Stryker, 1978*). Thus, a more complete picture of binocular combination of monocular responses shall consider the potential role of eye preference. This forms the basis of the current study, aiming to provide a more detailed understanding of how V1 neurons with varying eye preferences contribute to binocular vision.

Previous neurophysiological recording studies have revealed that, overall, the binocular responses of macaque V1 neurons are lower than the arithmetic sum of their respective monocular responses to the left and right eyes (*Prince et al., 2002*; *Mitchell et al., 2022*), but are stronger than the monocular responses when the neurons' preferred eye is stimulated (*Mitchell et al., 2022*). Furthermore, there is evidence suggesting that neurons' eye preferences play important functional roles. In a study by *Dougherty et al., 2019*, it was reported that the responses of monocular neurons are more likely to be suppressed than facilitated by binocular stimulation. As our forthcoming results will indicate, this conclusion holds true when considering the monocular baseline as either the sum of the monocular responses or the monocular responses of either eye. *Dougherty et al., 2019* also documented similar responses to monocular stimulation between neurons with monocular and binocular preferences, as well as comparable responses of binocular neurons to monocular and binocular stimulations, which are not supported by our data. In addition, *Mitchell et al., 2023* reported that the binocular combination of monocular stimuli with different contrasts is also influenced by neurons' eye preferences.

A more comprehensive understanding of binocular combination of monocular responses, as well as the influences of eye preferences, can be obtained from large samples of neurons in neighboring ocular dominance columns. This approach allows for quantitative descriptions with sufficient statistical power and analysis through computational modeling. Two-photon calcium imaging is well-suited for this task as it can record thousands of neurons simultaneously at single-neuron resolution. In this study, we used a two-photon calcium imaging setup that was custom-tailored for recording in awake macaques (*Li et al., 2017*) to measure the responses of V1 superficial-layer neurons to monocular and binocular stimulations. Additionally, we propose a neurally realistic binocular combination model that takes into account ocular dominance-dependent interocular divisive inhibition and binocular summation to interpret the gathered data.

## Results

We recorded responses of V1 superficial-layer neurons to monocular (contralateral and ipsilateral) and binocular stimulations in three awake, fixating macaques. The stimulus was a high-contrast (0.9) Gabor grating presented at various orientations and spatial frequencies (SFs). Recordings were performed within the same response field of view (FOV) at two cortical depths in first two monkeys (MA & MB), and at a single depth in the third monkey (MC) as the first two monkeys had displayed similar results at two depths (*Figure 1A*). A total of 10,168 neurons were identified through imaging processing, with 9390 (92.3%) tuned to orientation, SF, or both. Results from these orientation- and/or SF-tuned neurons were used in subsequent data analysis.

V1 superficial-layer neurons exhibited various degrees of eye preferences, consistent with the classical findings of *Hubel and Wiesel, 1968*. An ocular dominance index (ODI) was calculated to characterize each neuron's eye preference: ODI = $(R_i – R_c)/(R_i + R_c)$, in which $R_i$ and $R_c$ represented the neuron's respective peak responses to ipsilateral and contralateral stimulations on the basis of data fitting (see Materials and methods). An ODI of –1 or 1 would indicate complete contralateral or ipsilateral eye preference, respectively, while an ODI of 0 would indicate equal preferences to both eyes. The ocular dominance functional maps at single-neuron resolution, especially those of Monkeys A and C, revealed regions of neurons preferring either the contralateral (blue) or the ipsilateral eye (red), along with transitional zones where neurons showed preferences for both eyes (white) (*Figure 1B*). The ocular dominance maps were similar at two cortical depths in Monkeys A and B, suggesting the presence of ocular dominance columns (*Figure 1B*). The frequency distributions of ODIs suggest more binocular neurons than monocular neurons in V1 superficial layers (*Figure 1C*), similar to the normal distribution of ocularity index in *Dougherty et al., 2019*.

The responses of individual neurons were plotted against the ocular dominance index (ODI) when monocular stimulation was presented through the contralateral eye (*Figure 2A*) or the ipsilateral eye (*Figure 2B*). As expected, neurons with more negative ODIs responded stronger when

Neuroscience

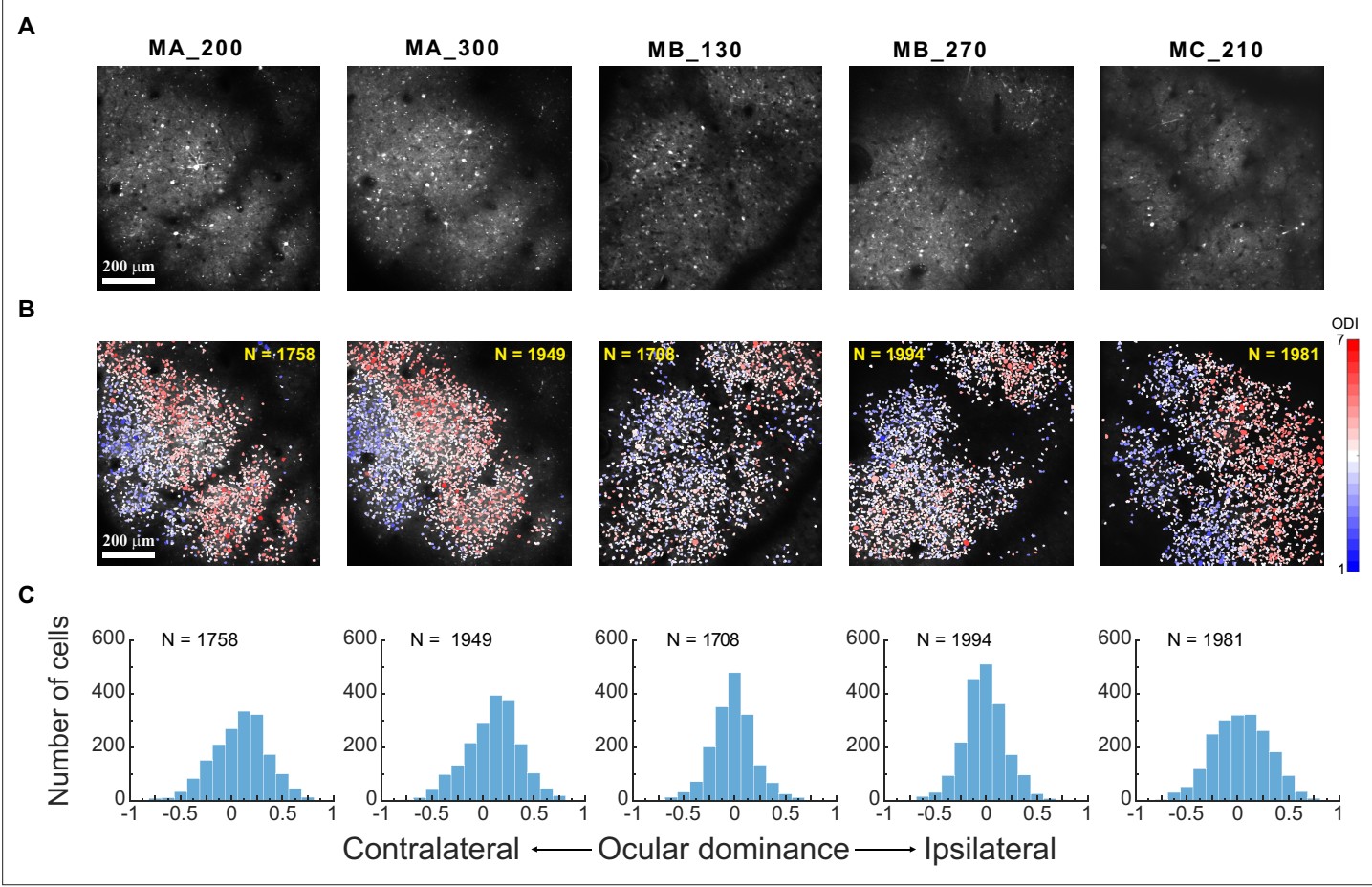

**Figure 1.** Eye preferences of V1 superficial-layer neurons in three macaques. (**A**) Two-photon imaging. Average two-photon images over a recording session for each response FOV. MA_200: Monkey A at a 200 µm cortical depth. (**B**) Ocular dominance functional maps of each FOV/depth at single-neuron resolution. (**C**) Frequency distributions of neurons of each FOV/depth as a function of ocular dominance index. Relevant data are provided in the source data file: *Figure 1—source data 1*.

The online version of this article includes the following source data for figure 1:

**Source data 1.** Source data of *Figure 1B-C*.

the contralateral eye was stimulated, and those with more positive ODIs responded stronger when the ipsilateral eye was stimulated. The responses declined as neurons became more binocular. Interestingly, when both eyes were stimulated, the response differences among neurons of different eye preferences became not obvious (*Figure 2C*).

These trends are more easily appreciated when comparing the differences between binocular and monocular responses of the same neurons for each FOV/depth in *Figure 2D*. The response changes varied among individual neurons - some showing response enhancement and some showing inhibition - likely reflecting the behaviors of tuned excitatory and inhibitory neurons, respectively (*Poggio and Fischer, 1977*). However, the overall response changes (represented by white dots connected by a black line, *Figure 2D and E*) are consistent among all five datasets. Specifically, under monocular stimulation, when only considering the neuronal responses to the preferred eye (i.e. a neuron's higher response to ipsilateral vs. contralateral stimulations), more monocular neurons (ODIs farther from 0) tended to exhibit stronger responses compared to more binocular neurons (ODIs closer to 0). However, under binocular stimulation, the overall responses of more monocular neurons were suppressed, and those of more binocular neurons were enhanced, by binocular stimulation. These trends are best appreciated in pooled data over five FOVs/depths (*Figure 2E*). Furthermore, linear regression verified a significant dependence of binocular modulation on absolute ODI (y=–0.36 x+0.14, p<0.001) (*Figure 2E*). This means that the responses of neurons with lower absolute ODI (i.e., binocular neurons)

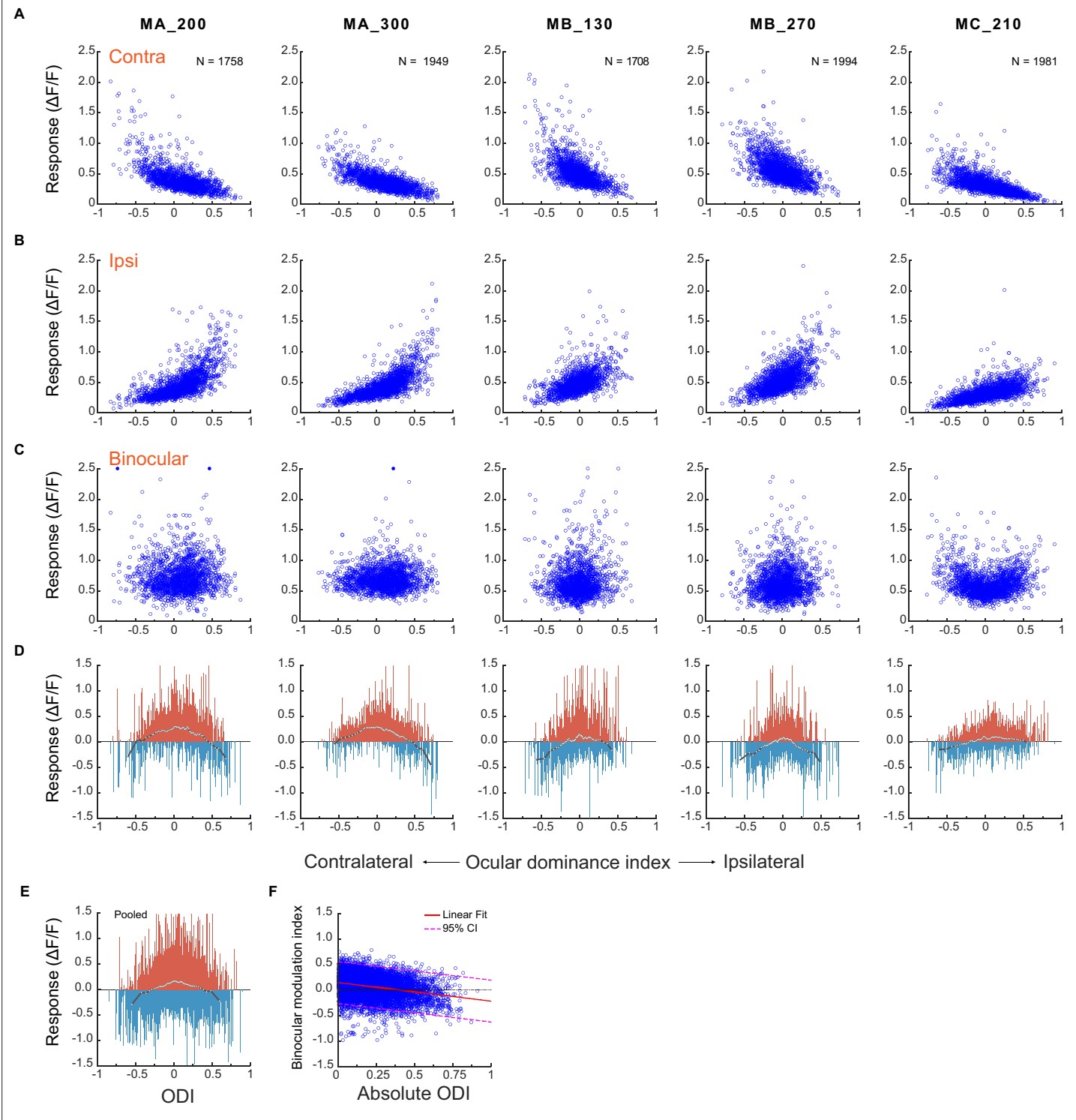

**Figure 2.** A comparison of neuronal responses to monocular and binocular stimulations. (**A**) Responses of individual neurons against their ocular dominance indices with contralateral stimulation. (**B**) Responses of the same neurons against their ocular dominance indices with ipsilateral stimulation. (**C**) Binocular responses of the same neurons. (**D**) The difference between binocular and monocular responses ($R_b$ – max($R_i$, $R_c$)). Each vertical line represents one neuron. To summarize the results, neurons of each FOV/depth are evenly divided into 60 bins in the order of the ocular dominance index. White dots represent the median responses of respective bins and are connected with a black line. (**E**) The differences between binocular and monocular responses of individual neurons pooled over five FOVs/depths. (**F**) Binocular modulation index as a function of absolute ODI and the linear

*Figure 2 continued on next page*

*Figure 2 continued*

fit. The binocular modulation index of each neuron was defined as ($R_b$ – max ($R_i$, $R_c$))/($R_b$ +max ($R_i$, $R_c$)). Relevant data are provided in the source data file: *Figure 2—source data 1*.

The online version of this article includes the following source data for figure 2:

**Source data 1.** Source data of *Figure 2*.

tended to be more enhanced, and responses of neurons with higher absolute ODI (i.e. monocular neurons) tended to be more suppressed.

## Modeling monocular and binocular responses

We used the following steps to develop a model that can account for the current monocular responses and their binocular combination. Monocular and binocular data of each FOV/depth, as well as the pooled data, were first normalized by the respective median of the binocular responses of all neurons in the same FOV/depth. The normalized data were then divided into 60 bins in the order of the ocular dominance index, and the median values of 60 bins were used for model fitting (*Figure 3A–C*).

## Monocular responses

First, a neuron's monocular responses to contralateral and ipsilateral stimulations were respectively described by a divisive gain control model:

$$R_c = \frac{w_c^{m_c} \cdot S_c}{k} \text{ and } R_i = \frac{w_i^{m_i} \cdot S_i}{k} \tag{1}$$

Here, $S_c$ and $S_i$ were stimulus contrasts, $w_c$ and $w_i$ were linear transformations of a neuron's ocular dominance index from [–1 1] to [0 1]: $w_c$ = (ODI +1)/2 and $w_i$ = 1 – $w_c$, $m_c$ and $m_i$ represented monocular nonlinearity, and k represented divisive gain control. Because $S_c$ and $S_i$ in our experiments were 0.9, which were about equal to 1 (full contrast) due to neuronal response saturation, the above equations were simplified as

$$R_c = \frac{w_c^{m_c}}{k} \text{ and } R_i = \frac{w_i^{m_i}}{k} \tag{2}$$

*Equation 2* was used to fit the binned median contralateral and ipsilateral data (*Figure 3A & B*), with the parameter k being equal for contralateral and ipsilateral responses. The fitting revealed that $m_c$ and $m_i$ were negative and close to –1, which resulted in a quick decline of $w_c^{m_c}$ and a quick increase of $w_i^{m_i}$ as a function of the ocular dominance index since $w_c$ and $w_i \in$ [0, 1]. The fit quality indices (*Busse et al., 2009*) ranged 0.92–0.94 for the contralateral condition (*Figure 3A*) and 0.87–0.93 for the ipsilateral condition (*Figure 3B*), suggesting adequate goodness of fit. The fitting parameters are listed in *Figure 3D*.

## Binocular responses

*Figure 2D–F* earlier has indicated that the overall neuronal responses to binocular stimulation change from suppression to enhancement as neurons' ocular dominance changes from monocular to binocular, which may reflect the ocular dominance-dependent net effect of interocular suppression and binocular summation. Therefore, we added interocular response suppression to *Equation 2* by letting monocular responses from each eye be further normalized by an interocular suppression factor $w_i^b$ or $w_c^b$ (*Equation 3*). In other words, the strength of interocular response suppression was decided by the linearly transformed ODI with a nonlinear exponent b. Finally, the normalized responses from two eyes were summed to simulate the binocular responses of neurons $R_b$, completing the model of binocular combination (*Equation 3*).

$$R_b = \frac{w_i^{m_i}}{k w_i^b} + \frac{w_c^{m_c}}{k w_c^b} \tag{3}$$

Although not shown in *Equation 3*, we also assumed that the nonlinear exponent b also depends on the contrast of the stimulus presented to the other eye (i.e., $S_c$ or $S_i$). Consequently, when $S_c$ or $S_i$

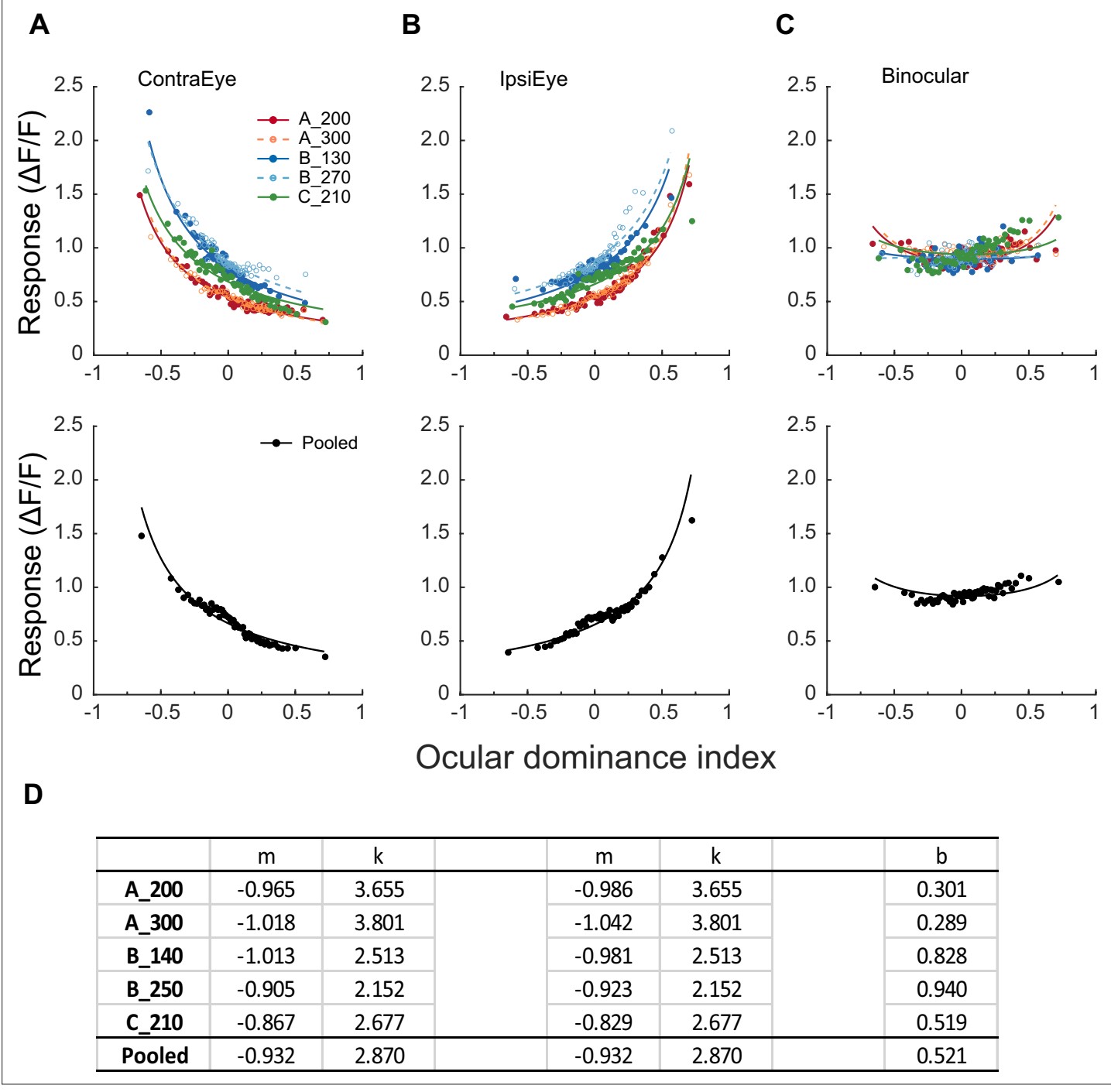

**Figure 3.** Modeling monocular and binocular responses. (**A**, **B**) Median neuronal responses to contralateral and ipsilateral stimulations as a function of ocular dominance index and respective data fitting with **Equation 2**. Neurons are evenly divided into 60 bins in the order of the ocular dominance index, with each bin containing 29–33 neurons that varied among different FOVs/depths (156 neurons for the pooled data). Each datum represents the median response of a bin. Free parameter k was kept equal during contralateral and ipsilateral data fitting. (**C**) Binocular responses as a function of ocular dominance index and data fitting with **Equation 3** for the same bins of neurons. During binocular data fitting, parameters k, $m_i$, and $m_c$ were inherited from monocular data fitting, and only b was a free-changing parameter. (**D**) The values of free parameters from monocular and binocular data fitting.

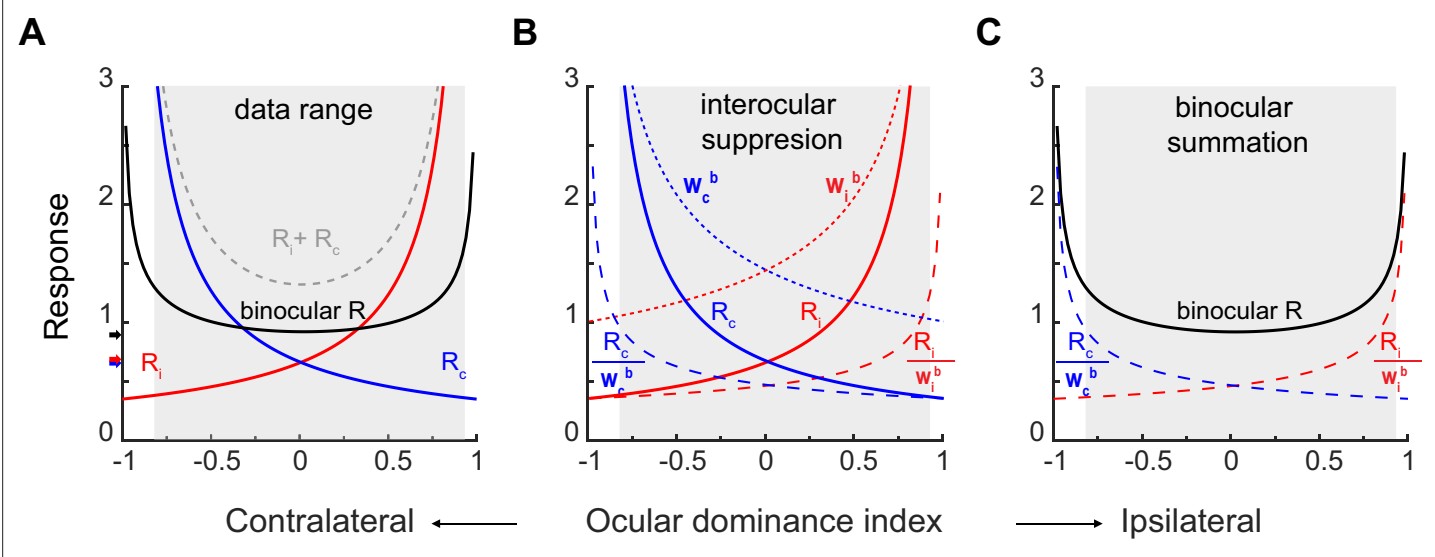

**Figure 4.** Monocular and binocular responses modeled with binocular suppression and binocular summation. (**A**) The solid blue, red, and black curves are respectively simulations of the contralateral, ipsilateral, and binocular responses on the basis of fitting of pooled data (**Figure 3D**). The grey dashed curve simulates binocular responses as the arithmetic sum of contralateral and ipsilateral responses. The higher branches of contralateral and ipsilateral response functions represent monocular responses with preferred eye stimulation. The black, blue, and red arrows indicate the median binocular, contralateral, and ipsilateral responses, respectively, from pooled data. The shadowed area indicates the region where actual neurons existed on the basis of the ocular dominance index. (**B**) Interocular suppression. The contralateral and ipsilateral responses ($R_c$ & $R_i$) are divided by respective interocular suppression factors $W_c^b$ and $W_i^b$ to produce interocular-suppressed responses $R_c/w_c^b$ and $R_i/w_i^b$. (**C**) Binocular summation. $R_c/w_c^b$ and $R_i/w_i^b$ are summed to produce the final binocular responses $R_b$.

= 0 under monocular stimulation, $R_c$ or $R_i$ = 0 (**Equation 1**), and interocular suppression $w_i^b$ or $wc_b$ = 1, so **Equation 3** changes back to **Equation 2**. It is only when $S_c$ and $S_i$ are equal and close to 1, as in the current study, that interocular suppression and binocular combination would be in the current **Equation 3** format.

When fitting binocular responses, the parameters $m_i$, $m_c$, and $k$ were inherited from earlier monocular data fitting and remained fixed. Only $b$ was allowed to change. Data fitting resulted in flat binocular response functions (**Figure 3C**) with satisfactory goodness-of-fit (fit quality index = 0.88–0.92).

The effects of interocular suppression and binocular summation in the model, as well as their contributions to the binocular response, may be better appreciated in **Figure 4**. **Figure 4A** uses **Equations 2 and 3** and the parameters from fitting of pooled data (**Figure 3D**) to simulate the contralateral (blue curve with label $R_c$), ipsilateral (red curve with label $R_i$), and binocular (black curve) response functions against the ocular dominance index. The arithmetic sum of contralateral and ipsilateral response functions was also simulated (grey dashed curve labeled $R_i + R_c$). In addition, neuronal responses to preferred eye stimulation would consist of the higher branches of contralateral and ipsilateral response functions. It is apparent that binocular responses cannot be explained by the sum of monocular responses, as binocular responses are substantially lower than the summed monocular responses for both monocular and binocular neurons. Nor can binocular responses be explained by the responses to the preferred eye, as binocular responses are also mostly lower than those to the preferred eye (the larger of the two monocular responses) for monocular neurons. Instead, the median of the binocular response function (black arrow by y-axis) in each data set is close to but still more or less higher than the median of the contralateral (blue arrow) or ipsilateral response function (red arrow), which is consistent with previous reports (**Prince et al., 2002**; **Mitchell et al., 2022**).

**Figure 4B** plots the interocular suppression factor $w_c^b$ (dotted blue curve) for the contralateral response $R_c$ (solid blue curve from 4 A), and $w_i^b$ (dotted red curve) for the ipsilateral response $R_i$ (solid red curve from 4 A). The interocular suppression factors $w_c^b$ and $w_i^b$ are larger with neurons that are more monocular than with neurons that are more binocular. The $R_c$ and $R_i$ are divided by the respective interocular suppression factors, producing the normalized contralateral responses (dashed blue curve labeled $R_c/w_c^b$) and ipsilateral responses (dashed red curve labeled $R_i/w_i^b$). These normalized curves

are lower than original monocular responses, especially for neurons that are more monocular (ODIs farther from 0), showing interocular suppression.

Then the normalized monocular curves are summed up in *Figure 4C*, which represents binocular summation and produces the final binocular responses ($R_b$). Therefore, as a result of combined interocular suppression and binocular summation, the final curve for binocular responses becomes nearly flat within the data range.

## Discussion

The current study compared the responses of large samples of V1 superficial-layer neurons to monocular and binocular stimulations in three macaques. The monocular response functions exhibited drastic changes as a function of the ocular dominance index, but binocular response functions remained largely flat, regardless of neurons' eye preferences. Modeling efforts indicated that when signals from two eyes are combined, interocular divisive suppression, which is more prominent with neurons preferring one eye, and binocular summation, which is more prominent with binocular neurons, together produce the nearly flat binocular response function within a broad range of ocular dominance indices. These findings imply that at least for neurons in superficial layers of V1, significant ocular dominance may stem from a release of interocular suppression during monocular stimulation, an unusual viewing scenario as our vision is typically binocular, rather than a lack of binocular combination of inputs from upstream monocular neurons.

We introduced this paper by citing the stable vision with monocular or binocular viewing. Relevant to this issue, in *Figure 4A* we observe that the median binocular neuronal responses is only marginally higher than those of monocular responses from each eye (black vs. red and blue arrows), consistent with earlier reports (*Prince et al., 2002*; *Mitchell et al., 2022*). The overall similarity between monocular and binocular responses likely plays a significant role in maintaining stable vision in both viewing conditions. What captivates us is the finding that seemingly very small response changes represent the net effects of profound binocular suppression with monocular neurons and facilitation with binocular neurons. Moreover, the slightly elevated binocular responses hint at possible further modulation by additional mechanisms to sustain stable vision.

In the study by *Dougherty et al., 2019*, they found that monocular and binocular neurons exhibit similar responses under monocular stimulation, with only monocular neurons, but not binocular neurons, being significantly suppressed by binocular stimulation. In contrast, our results reveal that monocular neurons show much stronger responses to stimulation in the preferred eye compared to binocular neurons under the same monocular stimulation (*Figure 2C*). Moreover, while we confirm the presence of interocular suppression in monocular neurons responding to the preferred eye as observed in *Dougherty et al., 2019*, our results also indicate enhanced responses in binocular neurons to binocular stimulation. The large diversity of binocular responses from neurons with similar eye preferences (*Figure 2C*) implies that obtaining accurate statistical estimates with small and potentially biased samples of neurons in electrode recording experiments may be challenging. In addition, it remains unclear whether the discrepancies observed are caused by differences in temporal resolutions of electrode recording and calcium imaging techniques. The results of *Dougherty et al., 2019* represent changes of neuronal spike activities over a period of approximately 50–200ms after stimulus onset, which may reflect the sustained neuronal responses to the stimulus and possible feedback signals. In contrast, calcium signals are much slower and capture aggregated neuronal responses over a longer period up to 1000ms in the current study, which should theoretically reduce through averaging, rather than exaggerate, the differences between monocular and binocular responses. However, we cannot rule out the possibility that neuronal response changes beyond 200ms may contribute to these discrepancies.

Binocular combination of monocular signals has been understood to involve both interocular suppression and binocular summation (*DeSliva and Bartley, 1930*; *Cohn and Lasley, 1976*; *Li and Atick, 1994*). Many more recent models have adopted divisive normalization to explain interocular suppression (e.g. *Cogan, 1987*; *Anderson and Movshon, 1989*; *Ding and Sperling, 2006*; *Moradi and Heeger, 2009*; *Huang et al., 2010*; *Ding et al., 2013*; *Mitchell et al., 2023*). Our modeling work suggests that a similar divisive interocular suppression and binocular summation model can effectively account for changes in neuronal responses under monocular and binocular stimulations, with the distinction that the divisive interocular suppression is additionally controlled by neurons' ocularity

preferences. The critical role of ocular dominance has been largely overlooked in extant binocular vision models to our knowledge, with exceptions that *Anderson and Movshon, 1989* in their model incorporating multiple ocular dominance channels to explain psychophysical adaptation data, and by *Mitchell et al., 2023* who showed that neurons' ocularity preference influences binocular combination of different contrasts from two eyes. We hope that our two-photon imaging results can be incorporated into future neuronally plausible models of binocular vision.

On the basis of current findings, future two-photon imaging work shall aim to compare neural responses to monocular and binocular stimulations with uneven effective stimulus contrasts due to physical contrast differences (*Anderson and Movshon, 1989*; *Mitchell et al., 2023*), monocular adaptation (*Anderson and Movshon, 1989*), short-term monocular deprivation (*Lunghi et al., 2011*), and the relevant roles of ocular dominance of individual neurons. These investigations would enhance the understanding of abnormal binocular vision in patients with strabismus and amblyopia. In addition, in our experiments, binocular stimuli were presented with zero disparity, which best affected the responses of neurons with zero-disparity tuning (including tuned excitatory and inhibitory neurons, *Figure 2*). A more realistic model of binocular combination also requires the consideration of neurons with other disparity-tuning profiles.

## Limitations of two-photon calcium imaging

While two-photon calcium imaging has the advantage of sampling a large number of neurons at cellular resolution with low sampling bias, it also has its known limitations that would position it as a complementary research tool to electrophysiological recording. For example, two-photon imaging can only sample neurons from superficial-layers, while binocular neurons also exist in deeper layers, and even neurons in the input layer are affected by feedback from downstream binocular neurons to exhibit binocular response properties (*Dougherty et al., 2019*). Furthermore, calcium signals are relatively slow and cannot capture the fast dynamics of neuronal responses. Consequently, to gain a more comprehensive understanding of the neuronal mechanisms involved in the binocular integration of monocular responses, combining both two-photon calcium imaging and electrophysiological recordings may offer a more holistic perspective.

In addition, it is important to consider that calcium signals may exaggerate the nonlinear properties of neurons. Although calcium signals indicated by GCaMP5, our preferred calcium indicator, displays a linear relationship to neuronal spike rates within a range of 10–150 Hz (*Li et al., 2017*), weaker and stronger signals out of this range are more nonlinear, and may appear poorer and stronger, respectively, than electrode-recorded effects. Consequently, the differences in population responses between monocular and binocular stimulations revealed by this study might be less pronounced.

## Materials and methods
### Monkey preparation

Monkey preparations were conducted following the methodology outlined in a previous study (*Guan et al., 2021*; *Ju et al., 2021*). Three rhesus monkeys (*Macaca mulatta*), aged 4–6 years, underwent two sequential surgeries under general anesthesia and strict sterile conditions. During the initial surgery, a 20 mm diameter craniotomy was performed on the skull over V1. The dura was opened, and multiple tracks of 100–150 nL AAV1.hSynap.GCaMP5G.WPRE.SV40 (AV-1-PV2478, titer 2.37e13 (GC/ml), Penn Vector Core) were pressure-injected at a depth of approximately 350 μm. Then the dura was sutured, the skull cap was re-attached using three titanium lugs and six screws, and the scalp was sutured. Following the surgery, the animal was returned to the cage and treated with injectable antibiotics (Ceftriaxone sodium, Youcare Pharmaceutical Group, China) for 1 week, along with postoperative analgesia. The second surgery took place 45 days later. A T-shaped steel frame was installed for head stabilization, and an optical window was inserted onto the cortical surface. Data collection could commence as early as one week following this procedure. More details about the preparation and surgical procedures can be found in *Li et al., 2017*. The procedures were approved by the Institutional Animal Care and Use Committee, Peking University.

## Behavioral task

After a ten-day recovery period following the second surgery, the monkeys were placed in primate chairs with head restraints. They were trained to maintain fixation on a small white spot (0.1°), with eye positions monitored by an ISCAN ETL-200 infrared eye-tracking system (ISCAN Inc) at a sampling rate of 120 Hz for Monkeys A and B, and an Eyelink-1000 (SR Research) at a sampling rate of 1000 Hz for Monkey C. During the experiment, trials in which the eye position deviated 1.5° or more from the fixation point before stimulus offset were excluded as ones with saccades and repeated. For the remaining trials, the eye positions were predominantly concentrated around the fixation point, with eye positions within 0.5° from the fixation point in over 95% of trials.

## Visual stimuli

For Monkeys A and B, visual stimuli were generated using a ViSaGe system (Cambridge Research Systems) and presented on a 21″ Sony G520 CRT monitor with a refresh rate of 80 Hz, a resolution of 1280 pixel ×960 pixel, and a pixel size of 0.31 mm × 0.31 mm. Due to space constraints, the viewing distance and the monitor position varied depending on the stimulus spatial frequency (30 cm for 0.25, 0.5, and 1 cpd, 60 cm for 2 cpd, and 120 cm for 4 and 8 cpd). For Monkey C, visual stimuli were created using Psychotoolbox 3 (*Pelli and Zhang, 1991*) and presented on a 27″ Acer XB271HU LCD monitor with a refresh rate of 80 Hz native, a resolution of 2560 pixel ×1,440 pixel native, and a pixel size of 0.23 mm × 0.23 mm. The viewing distance was 50 cm for lower frequencies (0.25–1 cpd) and 100 cm for higher frequencies (2–8 cpd). Both monitors had their screen luminance linearized by an 8-bit look-up table, and the mean luminance was approximately 47 cd/m$^2$.

A drifting square-wave grating with a spatial frequency of 4 cpd, a full contrast, a speed of 3 cycles/sec, a starting phase at 0°, and a size of 0.4° in diameter was initially used to determine the location, eccentricity (3.4° for Monkey A, 1.7° for Monkey B, and 1.1° for Monkey C), and size (0.8 - 1°) of the population receptive field associated with a recording field of view (FOV). Additionally, it was used to examine ocular dominance columns when presented monocularly to confirm the V1 location. This fast process involved a 4× objective lens mounted on a two-photon microscope and did not provide cell-specific information.

Neuronal responses were measured using a high-contrast (0.9) Gabor patch, which is a Gaussian-windowed sinusoidal grating, drifting at 2 cycles/sec in opposite directions perpendicular to the Gabor's orientation. The Gabor grating had a starting phase of 0° and varied at 12 orientations from 0° to 165° in 15° steps, along with 6 spatial frequencies ranging from 0.25 to 8 cpd in 1-octave steps.

In addition, three stimulus sizes (with constant stimulus centers) were used at each spatial frequency for two purposes. Firstly, our pilot measurements suggested very strong surround suppression with larger stimuli. Therefore, comparing responses to different stimulus sizes could help approximate the RF size of each neuron that produced maximal response and least surround suppression. Secondly, larger stimuli would have better chances to trigger neurons whose RF centers and the stimulus center were misaligned. It is worth noting that for additional neurons whose RFs had less overlap even with the largest stimuli used, they would have weaker and less orientation-tuned responses because of the Gaussian-blurred stimulus edge. These neurons would most likely be filtered out during our multiple steps of selection of orientation tuned neurons (see below).

Specifically, the stimulus sizes, represented by the σ of the Gaussian envelope of the Gabor, were 0.64 $\lambda$ and 0.85 $\lambda$ at all spatial frequencies, and was additionally smaller at 0.42 $\lambda$ when the SFs were 0.25–1 cpd, and larger at 1.06 $\lambda$ when the SFs were 2–8 cpd ($\lambda$: wavelength). Gabors at various SFs, if having the same σ in wavelength unit, would have the same number of cycles. Here at the smallest σ (0.42 $\lambda$), the Gabors still had sufficient number of cycles (frequency bandwidths = 1 octave; *Graham, 1989*), so that the actual stimulus spatial frequencies were precise at nominal values. In terms of visual angle, σ = 1.68°, 2.56°, and 3.36° at 0.25 cpd; 0.84°, 1.28°, and 1.68° at 0.5 cpd; 0.42°, 0.64°, and 0.85° at 1 cpd; 0.34°, 0.42°, and 0.53° at 2 cpd; 0.17°, 0.21°, and 0.26° at 4 cpd, and 0.08°, 0.11°, and 0.13° at 8 cpd, respectively.

Each stimulus was presented for 1000 ms, followed by an inter-stimulus interval (ISI) of 1500 ms, allowing sufficient time for the calcium signals to return to baseline levels (*Guan et al., 2020*). Each stimulus condition was repeated 12 times, with six repetitions for each opposite drift direction. When presenting a stimulus monocularly to one eye, the other eye was covered with a translucent eye patch to minimize short-term monocular deprivation. For Monkey A, binocular recordings preceded

monocular recordings on separate days. During monocular recordings, contralateral and ipsilateral stimulations alternated in blocks of trials, with at least a 10-min break in between, during which the eye patch was taken off. Recording at a specific viewing distance was completed with all trials at relevant SFs pseudo-randomly presented before proceeding to the next viewing distance. For Monkeys B and C, binocular and monocular recordings were mixed and completed in two daily sessions. At a specific viewing distance, all binocular trials at relevant SFs were carried out first, then contralateral and ipsilateral trials were completed in alternating blocks of trials with a at least 10 min eye-patch-off break in between. Again, recordings at a specific viewing distance were completed before proceeding to a different distance.

Each block of trials typically lasted 20–25 min, but for Monkeys A and B, certain blocks involving three SFs could extend up to 45 min. The strength of fluorescent signals (mean luminance of a small area) was continuously monitored and adjusted as needed to account for any drift in fluorescent signals. We compared the response ratios of the last two trials over the first two trials for each stimulus condition in these extended blocks with ipsilateral and contralateral stimulations. The respective mean ratios were 0.94 and 0.86, suggesting that the recorded neuronal responses remained largely stable over the extended blocks of trials.

## Two-photon calcium imaging

Two-photon calcium imaging was performed with a Prairie Ultima IV (In Vivo) two-photon microscope (Prairie Technologies) on Monkeys A and B, or a FENTOSmart two-photon microscope (Femtonics) on Monkey C, and a Ti:sapphire laser (Mai Tai eHP, Spectra Physics). GCaMP5 was chosen as the indicator of calcium signals because the fluorescence activities it expresses are linearly proportional to neuronal spike activities within a wide range of firing rates from 10 to 150 Hz (*Li et al., 2017*). One FOV of 850 x 850 µm$^2$ was selected from each animal and imaged using a 1000 nm femtosecond laser under a 16× objective lens (0.8 N.A., Nikon) at a resolution of 1.6 µm/pixel. A fast resonant scanning mode (32 frames per second) was chosen to obtain continuous images of neuronal activity (8 frames per second after averaging every 4 frames). Recordings were first performed at a shallower depth, and some neurons with high brightness or unique dendrite patterns were selected as landmarks. In the next daily session, the same FOV at the same depth was first located with the help of the landmarks, and the depth plane was then lowered if recordings were performed at a deeper depth (Monkeys A & B). Because of the time limit, recordings at a specific FOV/depth with monocular and binocular stimulations were completed in 2–3 consecutive daily sessions, but the same neurons could be precisely tracked over multiple recording sessions with the use of landmark cues.

## Imaging data analysis: initial screening of ROIs

Data were analyzed with customized MATLAB codes. A normalized cross-correlation based translation algorithm (source code provided in *Source code 1*) was used to reduce motion artifacts (*Li et al., 2017*). Then the fluorescence changes were associated with corresponding visual stimuli through the time sequence information recorded by Neural Signal Processor (Cerebus system, Blackrock Microsystem). By subtracting the mean of the 4 frames before stimuli onset (*F0*) from the average of the 6th-9th frames after stimuli onset (*F*) across 5 or 6 repeated trials for the same stimulus condition (same orientation, spatial frequency, size, and drifting direction), the differential image (*ΔF=F F0*) was obtained.

For a specific FOV at a specific recording depth, the regions of interest (ROIs) or possible cell bodies were decided through sequential analysis of 216 differential images in the order of spatial frequency (6), size (3), and orientation (12) (6x3 x 12=216). The first differential image was filtered with a band-pass Gaussian filter (size = 2–10 pixels), and connected subsets of pixels (>25 pixels, which would exclude smaller vertical neuropils) with average pixel value >3 standard deviations of the mean brightness were selected as ROIs. Then the areas of these ROIs were set to mean brightness in the next differential image before the bandpass filtering and thresholding were performed. This measure gradually reduced the standard deviations of differential images and facilitated detection of neurons with relatively low fluorescence responses. If a new ROI and an existing ROI from the previous differential image overlapped, the new ROI would be on its own if the overlapping area OA <1/4 ROI$_{new}$, discarded if 1/4 ROI$_{new}$ <OA < 3/4 ROI$_{new}$, and merged with the existing ROI if OA >3/4 ROI$_{new}$. The

merges would help smooth the contours of the final ROIs. This process went on through all differential images twice to select ROIs. Finally, the roundness for each ROI was calculated as:

$$Roundness = \frac{\sqrt{4\pi \times A}}{P}$$

where $A$ was the ROI's area, and $P$ was the perimeter. Only ROIs with roundness larger than 0.9, which would exclude horizontal neuropils, were selected for further analysis.

## Imaging data analysis: orientation tuning, SF tuning, and ocular dominance

The ratio of fluorescence change ($\Delta F/F0$) was calculated as a neuron's response to a specific stimulus condition. For a specific cell's response to a specific stimulus condition, the $F0_n$ of the n-th trial was the average of 4 frames before stimulus onset, and $F_n$ was the average of 5th-8th frames after stimulus onset. $F0_n$ was then averaged across 12 trials to obtain the baseline F0 for all 12 trials (for the purpose of reducing noises in the calculation of responses), and $\Delta F_n/F0 = (F_n-F0)/F0$ was taken as the neuron's response to this stimulus at this trial. The final response was averaged over 11 trials, excluding the 12th trial that showed the weakest and often negative response. For a small portion of neurons (e.g.,~3% in Monkeys A,~8% in monkey B, and ~2% in Monkey C) showing direction selectivity as their responses to two opposite drifting directions differed significantly ($P<0.05$, Friedman test), the 6 trials at the preferred direction was considered for calculations of $\Delta F_n/F0$ as the cell's responses to a particular stimulus. F0 was still averaged over 12 trials at two opposite directions.

Several steps were then taken to determine whether a neuron was tuned to orientation and/or spatial frequency, and if so, its ocular dominance index. For each monocular condition, first the orientation, SF, and size ($\sigma$) producing the maximal response among all conditions were selected. Then responses to other 11 orientations and 5 SFs were decided at the selected SF and size. Second, to select orientation and/or SF tuned neurons, a non-parametric Friedman test was performed to test whether a neuron's responses at 12 orientations or 6 SFs were significantly different from each other at least under one monocular stimulation condition. To reduce Type-I errors, the significance level was set at $\alpha=0.01$. Third, for those showing significant orientation differences, the trial-based orientation responses of each neuron were fitted with a Gaussian model with a MATLAB nonlinear least-squares function: lsqnonlin:

$$R(\theta) = a_1 2^{-\left(\frac{\theta - \theta_0}{\sigma}\right)^2} + b$$

where $R(\theta)$ was the response at orientation $\theta$, free parameters $a_1$, $\theta_0$, $\sigma$, and $b$ were the amplitude, peak orientation, standard deviation of the Gaussian function (and half width at half height), and minimal response of the neuron, respectively. Only neurons with goodness of fit $R^2 >0.5$ at least under one stimulation condition were finally selected as orientation-tuned neurons. Fourth, for those showing significant SF difference, the trial-based SF responses of each neuron were further fitted with a Difference-of-Gaussian model.

$$R(sf) = a_1 e^{-\left(\frac{sf}{\sigma_1}\right)^2} - a_2 e^{-\left(\frac{sf}{\sigma_2}\right)^2} + b$$

where $R(sf)$ was a neuron's response at spatial frequency sf, free parameters $a_1$, $\sigma_1$, $a_2$, and $\sigma_2$ were amplitudes and standard deviations of two Gaussians, respectively, and $b$ was the minimal response among 6 spatial frequencies. Only those with goodness of fit $R^2 >0.5$ at least under one monocular stimulation condition were selected as SF tuned neurons.

The ocular dominance index (ODI) was calculated to characterize each orientation and/or SF tuned neuron's eye preference: $ODI = (R_i - R_c)/(R_i + R_c)$, in which $R_i$ and $R_c$ were the neuron's respective peak responses at the best orientation and SF to ipsilateral and contralateral stimulations on the basis of data fitting. Here $ODI = -1$ and 1 would indicate complete contralateral and ipsilateral eye preferences, respectively, and $ODI = 0$ would indicate equal preference to both eyes.

## Model fitting

Monocular and binocular data in *Figure 4* were fitted by *Equations 2 and 3*, respectively. The goodness-of-fit was indicated by a fit quality index q with a range of 0–1, which was the root mean square deviation between the observed responses and the model normalized by the observed response mean (*Busse et al., 2009*):

$$q = 1 - \frac{\sqrt{\frac{\sum_{i=1}^{n}(r_i - m_i)^2}{n}}}{\bar{r}}$$

where *i* was the $i_{th}$ bin, *r* was the median response of a specific bin, and m was the corresponding model prediction.

## Acknowledgements

This study was supported by a STI2030-Major Projects grant (2022ZD0204600), Natural Science Foundation of China grants (31230030 and 31730109), and funds from Peking-Tsinghua Center for Life Sciences, Peking University. We thank Jian Ding and Dennis Levi at UC Berkeley and Si Wu at Peking University for their comments during the writing of the manuscript.

## Additional information

### Funding

| Funder | Grant reference number | Author |
|---|---|---|
| Ministry of Science and Technology of the People's Republic of China | 2022ZD0204600 | Shi-Ming Tang |
| Natural Science Foundation of China | 31230030 | Cong Yu |
| Natural Science Foundation of China | 31730109 | Shi-Ming Tang |
| Peking University | | Shi-Ming Tang Cong Yu |

The funders had no role in study design, data collection and interpretation, or the decision to submit the work for publication.

### Author contributions

Sheng-Hui Zhang, Data curation, Formal analysis, Investigation, Writing - original draft; Xing-Nan Zhao, Data curation, Formal analysis, Investigation; Dan-Qing Jiang, Data curation; Shi-Ming Tang, Conceptualization, Funding acquisition, Methodology, Writing – review and editing; Cong Yu, Conceptualization, Formal analysis, Funding acquisition, Writing - original draft, Writing – review and editing

### Author ORCIDs

Shi-Ming Tang https://orcid.org/0000-0003-0294-3259
Cong Yu http://orcid.org/0000-0002-8453-6974

### Ethics

All experimental protocols were approved by the Peking University Animal Care and Use Committee (LSC-TangSM-5).

Reviewer #1 (Public Review): https://doi.org/10.7554/eLife.92839.3.sa1
Reviewer #2 (Public Review): https://doi.org/10.7554/eLife.92839.3.sa2
Reviewer #3 (Public Review): https://doi.org/10.7554/eLife.92839.3.sa3
Author response https://doi.org/10.7554/eLife.92839.3.sa4

# Additional files

## Supplementary files
- MDAR checklist
- Source code 1. Source code for removing motion artifacts.

## Data availability
*Figure 1—source data 1* and *Figure 2—source data 1* contain the numerical data used to generate the figures.

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
