## [Editor Report · eLife assessment]

Overall, the reviewers found the significance of the work **valuable** to the field of visual neuroscience, particularly given the large data set and strength of the method used that allowed for spatial analysis of neuronal responses in macaque V1. The evidence was deemed **compelling**, owing in part to the consistency of responses across animals and the fitness of modeling. The authors have addressed the major comments from reviewers and improved the manuscript through relation to prior literature and addressing specific limitations of the method used.

---

## [Referee Report · Reviewer #1 (Public Review)]

Summary:

Zhang et al., investigated the relationship between monocular and binocular responses of V1 superficial-layer neurons using two-photon calcium imaging. They found a strong relationship in their data: neurons that exhibited a greater preference for one eye or the other (high ocular dominance) were more likely to be suppressed under binocular stimulation, whereas neurons that are more equivalently driven by each other (low ocular dominance) were more likely to be enhanced by binocular stimulation. This result chiefly demonstrates the relationship between ocular dominance and binocular responses in V1, corroborating what has been shown previously using electrophysiological techniques with now much finer spatial resolution. The binocular responses were well-fitted by a model that institutes divisive normalization between the eyes that accounts for both the suppression and enhancement phenomena observed in the subpopulation of binocular neurons. In so doing, the authors reify the importance of incorporating ocular dominance in computational models of binocular combination.

The conclusions of this paper are well supported by the data. The authors deftly contextualize these important findings in the literature while also acknowledging the limitations of the methodology employed. Future work would do well to combine the spatial power of 2P imaging with the temporal power of electrophysiology to assess ocular dominance-dependent binocular combination across the V1 laminar microcircuit.

Strengths:

The two-photon imaging technique used to resolve the activity of individual neurons within intact brain tissue grants a host of advantages. Foremost, two-photon imaging confers considerably high spatial resolution. As a result, the authors were able to sample and analyze the activity from thousands of verified superficial-layer V1 neurons. The animal model used, awake macaques, is also highly relevant for the study of binocular combination. Macaques, like humans, are binocular animals, meaning they have forward-facing eyes that confer overlapping visual fields. Importantly, macaque V1 is organized into cortical columns that process specific visual features from the separate eyes just like in humans. In combination with a powerful imaging technique, this allowed the authors to evaluate the monocular and binocular response profiles of V1 neurons that are situated within neighboring ocular dominance columns, a novel feat. To this aim, the approach was well-executed and should instill confidence in the notion that V1 neurons combine monocular information in a manner that is dependent on the strength of their ocular dominance.

Weaknesses:

This study suffers no major weaknesses. The authors address the limitations of the methodology and have calibrated the interpretations accordingly.

---

## [Referee Report · Reviewer #2 (Public Review)]

Summary:

This study examines the pattern of responses produced by the combination of left-eye and right-eye signals in V1. For this, they used calcium imaging of neurons in V1 of awake, fixating monkeys. They take advantage of calcium imaging, which yields large populations of neurons in each field of view. With their data set, they observe how response magnitude relates to ocular dominance across the entire population. They analyze carefully how the relationship changed as the visual stimulus switched from contra-eye only, ipsi-eye only, and binocular. As expected, the contra-eye dominated neurons responded strongly with a contra-eye only stimulus. The ipsi-eye dominated neurons responded strongly with an ipsi-eye only stimulus. The surprise was responses to a binocular stimulus. The responses were similarly weak across the entire population, regardless of each neuron's ocular dominance. They conclude that this pattern of responses could be explained by interocular divisive normalization, followed by binocular summation.

Strengths:

A major strength of this work is that the model-fitting was done on a large population of simultaneously recorded neurons. This approach is an advancement over previous work, which did model-fitting on individual neurons. The fitted model in the manuscript represents the pattern observed across the large population in V1, and washes out any particular property of individual neurons. Given the large neuronal population from which the conclusion was drawn, the authors provide solid evidence supporting their conclusion. They also observed consistency across 5 field of views.

The experiments were designed and executed appropriately to test their hypothesis. Their data support their conclusion.

Weaknesses:

The nonlinear interocular suppression found in this study, could potentially be partially exaggerated by the nonlinear properties of calcium signals. One of the authors of this study has previously reported that the particular GCaMP used in this study has a nice proportional relationship with firing rate of a neuron. So the concern of exaggeration probably does not apply to this particular study. The concern would apply to others who try similar measurements with other versions of GCaMP.

The implication of their finding is that strong ocular dominance is the result of release from interocular suppression by a monocular stimulus, rather than the lack of binocular combination as many traditional studies have assumed. This could significantly advance our understanding of the binocular combination circuitry of V1. The entire population of neurons could be part of a binocular combination circuitry present in V1.

---

## [Referee Report · Reviewer #3 (Public Review)]

Summary

The authors have made simultaneous recordings of the responses of large numbers of neurons from the primary visual cortex of macaque monkeys using optical two-photon imaging of calcium signals from the superficial layers of the cortex. Recordings were made to compare the responses of the cortical neurons under normal binocular viewing of a flat screen with both eyes open and monocular viewing of the same screen with one eye's view blocked by a translucent filter. The screen displayed visual stimuli comprising small contrast patches of Gabor function distributions of luminance, a stimulus that is known to excite cortical neurons.

Strengths

This is an important data set, given the large number of neurons recorded. The authors present a simple model to explain binocular combination of neuronal signals from the right and left eyes. The work advances the use of two-photon imaging in the cerebral neocortex. The research design adds valuable information to our understanding of the organization of binocular vision in macaque monkeys, which are the only realistic animal model of human vision for the study of binocular interactions.

Limitations and Weaknesses

(1) Given that these recordings are made optically, these results reflect primarily activations of neurons in the superficial layers of the cortex. This limitation arises from the usual constraints (depth of cortex, degree of myelination) on optical imaging in the macaque cortex. This means that the sample of neurons forming this data set is not fully representative of the population of binocular neurons within the visual cortex. This limitation is important in comparing the outcome of these experiments with the results from other studies of binocular combination, which have used single-electrode recording. Electrode recording will result in a sample of neurons that is drawn from many layers of the cortex, rather than just the superficial layers, noting that electrode recordings also carry different risks of sampling bias.

(2) Single neuron recording of binocular neurons in the primary visual cortex has shown that these neurons often have some spontaneous activity. Assessment of this spontaneous level of firing is important for accurate model fitting [1]. The present imaging approach works exclusively with differential measurements of neuronal signals, so assessment of the level of spontaneous activity is not feasible.

(3) The arrangements for visual stimulation and comparison of binocular and monocular responses mean that the stereoscopic disparity of the binocular stimuli is always at zero or close to zero. The consequence is that the experimental design does not test the cortical response over a range of different binocular depths.

The animal's fixation point is in the centre of a single display that is viewed binocularly. The fixation point is, by definition, at zero disparity.. Provided that the animals accurately converged their eyes on the binocular fixation point, then the disparity of the visual stimuli across the whole display will always be at or close to zero. However, we already know from earlier work that neurons in the visual cortex exhibit a range of selectivity for binocular disparity. Some neurons have their peak response at non-zero disparities, representing binocular depths nearer than the fixation depth or beyond it.

There are also other neurons whose response is maximally suppressed by disparities at the depth of the fixation point (so-called Tuned Inhibitory [TI] neurons). The simple model and analysis presented in the paper for the summation of monocular responses to predict binocular responses will perform adequately for neurons that are tuned to zero disparity, so-called tuned excitatory neurons [TE], but is necessarily compromised when applied to neurons that have other, different tuning profiles for binocular disparity. Specifically, when neurons are stimulated binocularly with a non-preferred disparity, the binocular response may be lower than the monocular response [2, 3]. The same limitation applies to another recent paper [4].

This more realistic view of binocular responses needs to be considered further to gain a full picture of the operation of the visual cortex in responding to binocular depth

Citations

1. Prince, S.J.D., Pointon, A.D., Cumming, B.G., and Parker, A.J., (2002). Quantitative analysis of the responses of V1 neurons to horizontal disparity in dynamic random-dot stereograms. Journal of Neurophysiology, 87: 191-208.

2. Prince, S.J.D., Cumming, B.G., and Parker, A.J., (2002). Range and mechanism of encoding of horizontal disparity in macaque V1. Journal of Neurophysiology, 87: 209-221.

3. Poggio, G.F. and Fischer, B., (1977). Binocular interaction and depth sensitivity in striate and prestriate cortex of behaving rhesus monkey. Journal of Neurophysiology, 40: 1392-1405 doi 10.1152/jn.1977.40.6.1392.

4. B. A. Mitchell, K. Dougherty, J. A. Westerberg, B. M. Carlson, L. Daumail, A. Maier, et al. (2022) Stimulating both eyes with matching stimuli enhances V1 responses.

iScience 2022 Vol. 25 Issue 5 DOI: 10.1016/j.isci.2022.104182

---

## [Author Response]

The following is the authors’ response to the original reviews.

**Public Reviews:**

**Reviewer #1 (Public Review):**
Summary:Zhang et al., investigated the relationship between monocular and binocular responses of V1 superficial-layer neurons using two-photon calcium imaging. They found a strong relationship in their data: neurons that exhibited a greater preference for one eye or the other (high ocular dominance) were more likely to be suppressed under binocular stimulation, whereas neurons that are more equivalently driven by each other (low ocular dominance) were more likely to be enhanced by binocular stimulation. This result chiefly demonstrates the relationship between ocular dominance and binocular responses in V1, corroborating what has been shown previously using electrophysiological techniques but now with greater spatial resolution (albeit less temporal resolution). The binocular responses were well-fitted by a model that institutes divisive normalization between the eyes that accounts for both the suppression and enhancement phenomena observed in the subpopulation of binocular neurons. In so doing, the authors reify the importance of incorporating ocular dominance in computational models of binocular combination.The conclusions of this paper are mostly well supported by the data, but there are some limitations of the methodology that need to be clarified, and an expansion of how the results relate to previous work would better contextualize these important findings in the literature.Strengths:The two-photon imaging technique used to resolve the activity of individual neurons within intact brain tissue grants a host of advantages. Foremost, two-photon imaging confers considerably high spatial resolution. As a result, the authors were able to sample and analyze the activity from thousands of verified superficial-layer V1 neurons. The animal model used, awake macaques, is also highly relevant for the study of binocular combination. Macaques, like humans, are binocular animals, meaning they have forward-facing eyes that confer overlapping visual fields. Importantly, macaque V1 is organized into cortical columns that process specific visual features from the separate eyes just like in humans. In combination with a powerful imaging technique, this allowed the authors to evaluate the monocular and binocular response profiles of V1 neurons that are situated within neighboring ocular dominance columns, a novel feat. To this aim, the approach was well-executed and should instill further confidence in the notion that V1 neurons combine monocular information in a manner that is dependent on the strength of their ocular dominance.Weaknesses:While two-photon imaging provides excellent spatial resolution, its temporal resolution is often lower compared to some other techniques, such as electrophysiology. This limits the ability to study the fast dynamics of neuronal activity, a well-understood trade-off of the method. The issue is more so that the authors draw comparisons to electrophysiological studies without explicit appreciation of the temporal difference between these techniques. In a similar vein, two-photon imaging is limited spatially in terms of cortical depth, preferentially sampling from neurons in layers 2/3. This limitation does not invalidate any of the interpretations but should be considered by readers, especially when making comparisons to previous electrophysiological reports using microelectrode linear arrays that sample from all cortical layers. Indeed, it is likely that a complete picture of early cortical binocular processing will require high spatial resolution (i.e., sampling from neurons in neighboring ocular dominance columns, from pia mater to white matter) at the biophysically relevant timescales (1ms resolution, capturing response dynamics over the full duration of the stimulus presentation, including the transient onset and steady-state periods).

To address the same concern from all three reviewers, we discussed the technical limitations of two photon calcium imaging at the end of Discussion, including limited imaging depth, low temporal resolution, and nonlinearity. The relevant texts are copied here:

(Ln 304) “Limitations of the current study

Although capable of sampling a large number of neurons at cellular resolution and with low sampling bias, two-photon calcium imaging has its known limitations that may better make it a complementary research tool to electrophysiological recordings.

For example, two-photon imaging can only sample neurons from superficial-layers, while binocular neurons also exist in deeper layers, and even neurons in the input layer are affected by feedback from downstream binocular neurons to exhibit binocular response properties (Dougherty, Cox, Westerberg, & Maier, 2019). Furthermore, calcium signals are relatively slow and cannot reveal the fast dynamics of neuronal responses. Due to these spatial and temporal limitations, a more complete picture of the neuronal mechanisms underlying binocular combination of monocular responses may come from studies using both technologies.

In addition, calcium signals may exaggerate the nonlinear properties of neurons. Although calcium signals indicated by GCaMP5, our favored choice of calcium indicator, displays a linear relationship to neuronal spike rates within a range of 10-150 Hz (Li, Liu, Jiang, Lee, & Tang, 2017), weak and strong signals out of this range are more nonlinear, and may appear poorer and stronger, respectively, than electrode-recorded effects. Consequently, the differences in population responses between monocular and binocular stimulations revealed by this study might be less pronounced.”

**(Recommendations For The Authors):**
Overall, my main suggestion for the authors to improve the paper is to revise some of the interpretations of their results in relation to previous research. The purpose of the present study was to illustrate a more complete picture of the binocular combination of monocular responses by taking into consideration the ocular dominance of V1 cells (lines 34-36). A study published earlier this year had an identical purpose (Mitchell et al., Current Biology, 2023) and arrived at a highly similar conclusion (and also applied divisive normalization to fit their data). I would ask that this paper be mentioned in the introduction and discussed.

The Mitchell et al 2023 paper is added to the Introduction and Discussion:

(Ln 50) “In addition (to the Dougherty et al 2019 paper from the same group), Mitchell, Carlson, Westerberg, Cox, and Maier (2023) reported that binocular combination of monocular stimuli with different contrasts is also affected by neurons’ eye preference.”

(Ln 286) “The critical roles of ocular dominance have been largely overlooked by extant binocular vision models to our knowledge, except that Anderson and Movshon (1989) demonstrated that a model consisting of multiple ocular dominance channels can better explain their psychophysical adaptation data, and that Mitchell et al. (2023) revealed that binocular combination of different contrasts presented to different eyes are affected by neurons’ ocularity preference.”

Nevertheless, the results of the present study are very valuable. They add substantial spatial resolution and sophisticated relational analysis of monocular and binocular responses that Mitchell et al., 2023 did not include. Therefore, my suggestion is to emphasize the advantages of two-photon imaging in the introduction, focusing on the ability to image neurons in neighboring ocular dominance columns. The rigorous modeling of the relationship between nearby neurons with a range of eye preferences, in tandem with the incredible yield of two-photon imaging, is what sets this paper apart from previous electrophysiological work.The finding that binocular responses were dependent on ocular dominance is largely consistent with previous electrophysiological results. However, there should be a paragraph in the discussion section that speaks to the limitations of comparing two-photon imaging data to electrophysiological data. Namely, there are two limitations:(1) These two techniques confer different temporal resolutions. It is conceivable that some of the electrophysiology relationships (for example, described by Dougherty et al., 2019) may be dependent on the temporal window over which the data was averaged, typically over 50-100ms around stimulus onset, or 100-250ms comprising the neurons' sustained response to the stimulus. This possible explanation of the difference in obtained results would be especially useful for the discussion paragraph starting at line 232. It would also be helpful to readers for there to be some mention of the advantage of having high temporal resolution (i.e., the benefits of electrophysiology) since (a) recent work has distinguished between sequential stages of binocular combination (Cox et al., 2019) and (b) modern models of V1 neurons emphasize recurrent feedback to explain V1 temporal dynamics (see Heeger et al., 2019; Rubin et al., 2015), which could prove to be relevant for combination of stimuli in the two eyes (Fleet et al., 1997).

Our discussion regarding the technical limitations of 2-p calcium imaging has been listed earlier. Specific to the Dougherty et 2019 paper, we added the following discussion to address the issue of temporal resolution difference between two technologies.

(Ln 266) “In addition, it is unclear whether the discrepancies are caused by different temporal resolutions of electrode recording and calcium imaging. The results of Dougherty et al. (2019) represent changes of neuronal spike activities over a period of approximately 50-200 ms after the stimulus onset, which may reflect the sustained neuronal responses to the stimulus and possible feedback signals. Calcium signals are much slower and indicative of the aggregated neuronal responses over a longer period (up to 1000 ms in the current study). They should have smeared, rather than exaggerated, the differences between monocular and binocular responses, although we cannot exclude the possibility that some neuronal response changes beyond 200 ms are responsible for the discrepancies.”

(2) The sample of V1 neurons in this study is limited to cells in the most superficial layers of the cortex (layers 2/3). This limitation is, of course, well understood, but it should be mentioned at least in the context of studying the formative mechanisms of binocular combination in V1 (since we know that binocular neurons also exist in layers 5/6, and there is now substantial evidence that even layer 4 neurons are not as "monocular" as we previously thought (Dougherty et al., 2019)).

See our discussion regarding the technical limitations of 2-p calcium imaging listed earlier.

In short, I believe the paper would be improved by (1) adding the above citations in the appropriate places, (2) acknowledging in the introduction that this question has been investigated electrophysiologically but emphasizing the advantages of two-photon imaging, and (3) adding a paragraph to the discussion section that discusses the temporal and spatial limitations when using two-photon imaging to study binocular combination, particularly when comparing the results to electrophysiology.
**Reviewer #2 (Public Review):**
Summary:This study examines the pattern of responses produced by the combination of left-eye and right-eye signals in V1. For this, they used calcium imaging of neurons in V1 of awake, fixating monkeys. They take advantage of calcium imaging, which yields large populations of neurons in each field of view. With their data set, they observe how response magnitude relates to ocular dominance across the entire population. They analyze carefully how the relationship changed as the visual stimulus switched from contra-eye only, ipsi-eye only, and binocular. As expected, the contra-eye-dominated neurons responded strongly with a contra-eye-only stimulus. The ipsi-eye-dominated neurons responded strongly with an ipsi-eye-only stimulus. The surprise was responses to a binocular stimulus. The responses were similarly weak across the entire population, regardless of each neuron's ocular dominance. They conclude that this pattern of responses could be explained by interocular divisive normalization, followed by binocular summation.Strengths:A major strength of this work is that the model-fitting was done on a large population of simultaneously recorded neurons. This approach is an advancement over previous work, which did model-fitting on individual neurons. The fitted model in the manuscript represents the pattern observed across the large population in V1, and washes out any particular property of individual neurons. Given the large neuronal population from which the conclusion was drawn, the authors provide solid evidence supporting their conclusion. They also observed consistency across 5 fields of view.The experiments were designed and executed appropriately to test their hypothesis. Their data support their conclusion.Weaknesses:One weakness of their study is that calcium signals can exaggerate the nonlinear properties of neurons. Calcium imaging renders poor responses poorer and strong responses stronger, compared to single-unit recording. In particular, the dramatic change in the population response between monocular stimulation and binocular stimulation could actually be less pronounced when measured with single-unit recording methods. This means their choice of recording method could have accidentally exaggerated the evidence of their finding.

We discussed the nonlinearity of calcium signals as part of the technical limitations of 2-p imaging calcium. The calcium indicator we use, GCaMP5, has a reasonable range of linear relationship with spike rates. But out of this range, the nonlinearity is indeed a concern.

(Ln 314) “In addition, calcium signals may exaggerate the nonlinear properties of neurons. Although signals indicated by GCaMP5, our favored choice of calcium indicator, displays a linear relationship to neuronal spike rate within a range of 10-150 Hz (Li et al., 2017), weak and strong signals out of this range are more nonlinear, and may appear poorer and stronger, respectively, than electrode-recorded effects. Consequently, the changes in population responses between monocular and binocular stimulations revealed by this study might be less pronounced.”

The implication of their finding is that strong ocular dominance is the result of release from interocular suppression by a monocular stimulus, rather than the lack of binocular combination as many traditional studies have assumed. This could significantly advance our understanding of the binocular combination circuitry of V1. The entire population of neurons could be part of a binocular combination circuitry present in V1.

This is a very good insight. We added the following sentences to the end of the first paragraph of Discussion:

(Ln 242) “These findings implicate that at least for neurons in superficial layers of V1, significant ocular dominance may result from a release of interocular suppression during monocular stimulation, an unusual viewing condition as our vision is typically binocular, rather than a lack of binocular combination of inputs from upstream monocular neurons.”

**(Recommendations For The Authors):**
Line 150: "To model interocular response suppression, responses from each eye in Eq. 2 were further normalized by an interocular suppression factor wib or wcb," I recommend the authors improve their explanation of how they arrived at Eq. 3 from Eq. 2. As it stands, my impression is that they have one model for the responses to monocular stimulation, and another model for the responses to binocular stimulation. What I think is missing is that both equations are derived from the same model. Monocular stimulation is a situation in which the stimulus in one eye's contrast is zero. Could the authors clarify whether this situation produces an interocular suppression of zero, and how that leads to Eq. 2?

We rewrote the modeling part to show that Equations 1-3 are sequential steps of development for the same model. We also added a brief paragraph to discuss how Eq. 3 could lead to Eq. 2 under monocular viewing:

(Ln 166) “Although not shown in Eq. 3, we also assumed that the nonlinear exponent b also depends on the contrast of the stimulus presented to the other eye (i.e., Sc or Si). Consequently, when Sc or Si = 0 under monocular stimulation, Rc or Ri = 0 (Eq. 1), and interocular suppression wib or wcb = 1, so Eq. 3 changes back to Eq. 2. It is only when Sc and Si are equal and close to 1, as in the current study, that interocular suppression and binocular combination would be in the current Eq. 3 format.”

Line 225: "However, individually, compared to monocular responses, responses of monocular neurons more preferring the stimulated eye are actually suppressed, and only responses of binocular neurons are increased by binocular stimulation." This sentence is difficult to follow. I recommend the authors improve clarity by breaking up the sentence into several sentences. If I understand correctly, they summarize the pattern in the data that is indicative of interocular divisive normalization, i.e., their final conclusion.

This sentence no longer exists in the Discussion.

Line 426: "Third, for those showing significant orientation difference, the trial-based orientation responses of each neuron were fitted with a Gaussian model with a MATLAB nonlinear least squares function:" The choice of using a Gaussian function to fit orientation tuning was probably suboptimal. A Gaussian function provides an adequate fit only for neurons whose tuning is very sharp. The responses outside of the peak fall down to the baseline and the two ends meet. Otherwise, the two ends do not meet. An adequate fit would be achieved with a function of a circular variable, which wraps around 180 deg. I recommend using a Von Mises function for fitting orientation tuning.

We agree with the reviewer that the Von Mises function is more accurate than Gaussian for fitting orientation tuning functions. Indeed we are using it to fit orientation tuning of V4 neurons, many of which have two peaks. For the current V1 data, the differences between Von Mises and Gaussian fittings are very small, as shown in the orientation functional maps from three macaques below. Because we also use the same Gaussian fitting of orientation tuning in several published and current under-review papers, we prefer to keep the Gaussian fitting results in the manuscript.

**Author response image 1. sa4fig1:** 

**Reviewer #3 (Public Review):**
The authors have made simultaneous recordings of the responses of large numbers of neurons from the primary visual cortex using optical two-photon imaging of calcium signals from the superficial layers of the cortex. Recordings were made to compare the responses of the cortical neurons under normal binocular viewing of a flat screen with both eyes open and monocular viewing of the same screen with one eye's view blocked by a translucent filter. The screen displayed visual stimuli comprising small contrast patches of Gabor function distributions of luminance, a stimulus that is known to excite cortical neurons.This is an important data set, given the large numbers of neurons recorded. The authors present a simple model to explain the binocular combination of neuronal signals from the right and left eyes.The limitations of the paper as written are as follows. These points can be addressed with some additional analysis and rewriting of sections of the paper. No new experimental data need to be collected.(1) The authors should acknowledge the fact that these recordings arise from neurons in the superficial layers of the cortex. This limitation arises from the usual constraints on optical imaging in the macaque cortex. This means that the sample of neurons forming this data set is not fully representative of the population of binocular neurons within the visual cortex. This limitation is important in comparing the outcome of these experiments with the results from other studies of binocular combination, which have used single-electrode recording. Electrode recording will result in a sample of neurons that is drawn from many layers of the cortex, rather than just the superficial layers.

See our discussion regarding the technical limitations of 2-p calcium imaging listed earlier.

(2) Single-neuron recording of binocular neurons in the primary visual cortex has shown that these neurons often have some spontaneous activity. Assessment of this spontaneous level of firing is important for accurate model fitting [1]. The paper here should discuss the level of spontaneous neuronal firing and its potential significance.

We have noticed previously that at non-optimal spatial frequencies, calcium responses to a moving Gabor grating are close to zero (Guan et al., Prog Neurobiology, 2021, Fig. 1B), but we cannot tell whether this is due to calcium response nonlinearity, or a close-to-zero level of spontaneous neuronal activity. Prince et al (2002) reported low spontaneous responses of V1 neurons with moving grating stimuli (e.g., about 3 spikes/sec in one exemplar neuron, their Fig. 1B), so this appears not a big effect. In our data fitting, we do have an orientation-unspecific component in the Gaussian model, which represents the neuronal response at a non-preferred orientation, but not necessarily the spontaneous activity.

(3) The arrangements for visual stimulation and comparison of binocular and monocular responses mean that the stereoscopic disparity of the binocular stimuli is always at zero or close to zero. The animal's fixation point is in the centre of a single display that is viewed binocularly. The fixation point is, by definition, at zero disparity. The other points on the flat display are also at zero disparity or very close to zero because they lie in the same depth plane. There will be some small deviations from exactly zero because the geometry of the viewing arrangements results in the extremities of the display being at a slightly different distance than the centre. Therefore, the visual stimulation used to test the binocular condition is always at zero disparity, with a slight deviation from zero at the edges of the display, and never changes. [There is a detail that can be ignored. The experimenters tested neurons with visual stimulation at different real distances from the eyes, but this is not relevant here. Provided the animals accurately converged their eyes on the provided binocular fixation point, then the disparity of the visual stimuli will always be at or close to zero, regardless of viewing distance in these circumstances.] However, we already know from earlier work that neurons in the visual cortex exhibit a range of selectivity for binocular disparity. Some neurons have their peak response at non-zero disparities, representing binocular depths nearer than the fixation depth or beyond it. The response of other neurons is maximally suppressed by disparities at the depth of the fixation point (so-called Tuned Inhibitory [TI] neurons). The simple model and analysis presented in the paper for the summation of monocular responses to predict binocular responses will perform adequately for neurons that are tuned to zero disparity, so-called tuned excitatory neurons [TE], but is necessarily compromised when applied to neurons that have other, different tuning profiles. Specifically, when neurons are stimulated binocularly with a non-preferred disparity, the binocular response may be lower than the monocular response[2, 3]. This more realistic view of binocular responses needs to be considered by the authors and integrated into their modelling.

We agree and include the following texts when discussing the future work:

(Ln 298) “In addition, in our experiments, binocular stimuli were presented with zero disparity, which best triggered the responses of neurons with zero-disparity tuning. A more realistic model of binocular combination also requires the consideration of neurons with other disparity-tuning profiles.”

(4) The data in the paper show some features that have been reported before but are not captured by the model. Notably for neurons with extreme values of ocular dominance, the binocular response is typically less than the larger of the two monocular responses. This is apparent in the row of plots in Figure 2D from individual animals and in the pooled data in Figure 2E. Responses of this type are characteristic of tuned inhibitory [TI] neurons[2]. It is not immediately clear why this feature of the data does not appear in the summary and analysis in Figure 3.

This difference is indeed captured by the model, which can be more easily appreciated in Fig. 4A where monocular and binocular model simulations are plotted in the same panel. In the text, we also wrote: (Ln 195) “It is apparent that binocular responses cannot be explained by the sum of monocular responses, as binocular responses are substantially lower than the summed monocular responses for both monocular and binocular neurons. Nor can binocular responses be explained by the responses to the preferred eye, as binocular responses are also lower than those to the preferred eye (the larger of the two monocular responses) for monocular neurons.”

The paper text states that the responses were "first normalized by the median of the binocular responses". This will certainly get rid of this characteristic of the data, but this step needs better justification, or an amendment to the main analysis is needed.

The relevant sentence has been rewritten as “Monocular and binocular data of each FOV/depth, as well as the pooled data, were first normalized by the respective median of the binocular responses of all neurons in the same FOV/depth.” This normalization would render the overall binocular responses to be around unity, for the purpose of facilitating comparisons among all FOV/depth, but it would not affect the overall characteristic of the data.

In the present form, the model and analysis do not appear to fit the data in Figure 2 as accurately as needed.

Thanks for pointing out the problem, as data fitting for FOV C_270 and the pooled data were especially inaccurate. The issue has been mostly fixed when each datum was weighted by its standard deviation (please see the updated Fig. 3).